# Influence of Cluster-Situated Regulator PteF in Filipin Biosynthetic Cluster on Avermectin Biosynthesis in *Streptomyces avermitilis*

**DOI:** 10.3390/biology13050344

**Published:** 2024-05-15

**Authors:** Guozhong Du, Xue Yang, Zhengxiong Wu, Minghui Pan, Zhuoxu Dong, Yanyan Zhang, Wensheng Xiang, Shanshan Li

**Affiliations:** 1State Key Laboratory for Biology of Plant Diseases and Insect Pests, Institute of Plant Protection, Chinese Academy of Agricultural Sciences, No. 2 Yuanmingyuan West Road, Haidian District, Beijing 100193, China; dgz1005@163.com (G.D.); 82101211024@caas.cn (X.Y.); 17513232046@163.com (Z.W.); mhlu@163.com (M.P.); d15945317995@163.com (Z.D.); yyzhang@ippcaas.cn (Y.Z.); 2School of Life Science, Northeast Agricultural University, No. 59 Mucai Street, Xiangfang District, Harbin 150030, China

**Keywords:** avermectin, filipin, regulator, *Streptomyces avermitilis*, transcriptome

## Abstract

**Simple Summary:**

Crosstalk regulation is a common regulatory phenomenon in *Streptomyces*, typically mediated by regulatory factors within gene clusters. Utilizing these regulatory factors to enhance the production of target compounds represents an important approach. This study focuses on two regulatory factors, PteF and PteR, within the filipin gene cluster in *Streptomyces avermitilis*, investigating their effects on both avermectin production and overall cellular metabolism. The findings provide theoretical groundwork for constructing high-yielding engineered strains of avermectins and provide new insights into the role of PteF in the biosynthesis of avermectins and its impact on cellular metabolic processes.

**Abstract:**

Crosstalk regulation is widespread in *Streptomyces* species. Elucidating the influence of a specific regulator on target biosynthetic gene clusters (BGCs) and cell metabolism is crucial for strain improvement through regulatory protein engineering. PteF and PteR are two regulators that control the biosynthesis of filipin, which competes for building blocks with avermectins in *Streptomyces avermitilis*. However, little is known about the effects of PteF and PteR on avermectin biosynthesis. In this study, we investigated their impact on avermectin biosynthesis and global cell metabolism. The deletion of *pteF* resulted in a 55.49% avermectin titer improvement, which was 23.08% higher than that observed from *pteR* deletion, suggesting that PteF plays a more significant role in regulating avermectin biosynthesis, while PteF hardly influences the transcription level of genes in avermectin and other polyketide BGCs. Transcriptome data revealed that PteF exhibited a global regulatory effect. Avermectin production enhancement could be attributed to the repression of the tricarboxylic acid cycle and fatty acid biosynthetic pathway, as well as the enhancement of pathways supplying acyl-CoA precursors. These findings provide new insights into the role of PteF on avermectin biosynthesis and cell metabolism, offering important clues for designing and building efficient metabolic pathways to develop high-yield avermectin-producing strains.

## 1. Introduction

Avermectins are a class of 16-membered ring macrolide compounds, which currently represent the largest group of green natural-product pesticides produced and utilized globally owing to their outstanding properties, such as low dosage, long action time, low toxicity to mammals and easy decomposition [1]. *Streptomyces avermitilis* is the most significant producer of avermectins [2,3]. The titer of avermectin B_1a_ was improved from 0.009 g/L to 0.5 g/L by Merck using random mutation techniques, such as ultraviolet, methyl methanesulfonate and N-methyl-N′-nitro-N-nitrosoguanidine [4]. Further, with the rapid development of omics analytical technologies, high-yield avermectin producers have been engineered via different metabolic engineering strategies. For instance, the titer of avermectin B_1a_ of *S. avermitilis* has been enhanced through the manipulation of various transcription regulators, such as TetR family regulators SAV151 [5] and AveT [6], the MarR family regulator SAV4189 [7] and the sigma factor σ*^hrdB^* [8]. Titer improvement could also be achieved by engineering transporters for carbon substrate uptake (e.g., *malEFG*, TP2, TP5 and TP6568) and putative product efflux (e.g., AvtAB and MiltAB2) [9,10,11,12,13]. Moreover, the titer of avermectin B_1a_ was boosted by approximately 50% to 9.31 g/L using a triacylglycerol dynamic degradation strategy in our previous work [14], and Hao et al. enhanced the titer of avermectin B_1a_ to 9.61 g/L through a combination engineering strategy [15]. To date, the titer of avermectin B_1a_ has been increased over 1000-fold since its first discovery. Nevertheless, understanding the influence of various regulatory proteins on avermectin biosynthesis in *S. avermitilis* is still the tip of the iceberg, which impedes us from unlocking the new targets for further strain improvement and titer enhancement.

The genome of *Streptomyces* generally contains more than 20 secondary metabolite biosynthetic gene clusters (BGCs), including polyketides, non-ribosomal peptides, terpenes, ribosomally synthesized and post-translationally modified peptides, and so on [2]. The biosynthesis of secondary metabolites is tightly restricted by complex regulatory and metabolic networks, including cross-regulation and metabolic flux competition [16,17]. The production enhancement of target chemicals could be achieved via the inactivation of competitive BGCs [18], whereas its effectiveness is not always guaranteed due to cross-regulation. For example, we discovered that the *Streptomyces* antibiotic regulatory protein (SARP) family regulator protein KelR controls the biosynthesis of a yellow polyketide compound, KEL, as well as our target chemicals, milbemycins, in *Streptomyes bingchenggensis* [19]. Similarly, TtnQ is also an essential regulator that activates the biosynthesis of both tautomycetin and spirotoamide in *Streptomyces griseochromogenes* [20]. Therefore, it is important to clarify whether or not a given regulator of the competitive BGC effectively possesses cross-regulatory effects on the target chemicals prior to regulatory protein engineering.

The wild-type strain *S. avermitilis* MA-4680 contains 38 BGCs in total, including 7 putative type I polyketide BGCs [21]. Avermectin, filipin and oligomycin can be detected among these compounds under conventional fermentation conditions [2]. Since the three BGCs share the same biosynthetic extender units of malonyl-CoA and methylmalonyl-CoA, they undergo metabolic competition while being simultaneously cross-regulated by different regulatory proteins. The cluster-situated regulator (CSR) AveR activates avermectin biosynthesis while inhibiting oligomycin biosynthesis [22], and another CSR, PteF, positively regulates the biosynthesis of both filipin and oligomycin [23]. However, apart from investigations into PteF’s impact on the transcription level of avermectin BGC (*ave*) [24], few studies have focused on the relationship between filipin BGC (*pte*) and *ave*, as well as the influence of both CSR proteins PteF and PteR on avermectin production in *S. avermitilis*. Therefore, an elucidation of the influence of PteF and PteR on avermectin biosynthesis might be promising for guiding the rational engineering of high-yield avermectin producers via regulatory engineering.

In this work, we investigate the influence of PteF and PteR on avermectin production in *S. avermitilis* S0 with two copies of *ave* and *pte*, and elucidate the cross-regulatory effect of PteF on *pte* and *ave*. We further elucidate the mechanism of PteF deletion on avermectin production enhancement based on transcriptome data analysis. This work expands our understanding of the influence of PteF on avermectin biosynthesis, as well as on cell metabolism in *S. avermitilis*, providing new insights and targets for further strain engineering for higher avermectin production.

## 2. Materials and Methods

### 2.1. Strains, Plasmids and Culture Conditions

All strains and plasmids used in this study are listed in Appendix A. *Escherichia coli* JM109 was used for plasmid cloning; *E. coli* ET12567/pUZ8002 served as a donor in intergeneric conjugations. *S. avermitilis* S0 is a derivative of *S. avermitilis* MA-4680 obtained via random mutagenesis, which has been deposited at the China General Microbiology Culture Collection Center (accession no. CGMCC 4.8011). *E. coli* was grown on Luria Bertani (LB) medium supplemented with antibiotics as required at 37 °C. For conjugations, *S. avermitilis* strains were grown on mannitol soya flour (MS) medium at 28 °C for 7 days [25]. For fermentations of *S. avermitilis* and their derivatives, seed cultures were prepared by inoculating one square centimeter of spores into 250 mL Erlenmeyer flasks, containing 25 mL of seed medium and cultivated at 28 °C for 40 h with a speed of 250 rpm. After that, 1.5 mL of seed culture was transferred into 250 mL Erlenmeyer flasks containing 25 mL of fermentation medium, and further cultured for 10 days. The seed medium and fermentation medium of *S. avermitilis* were described in previous studies [26].

### 2.2. Construction of In-Frame Deletion Mutant Strains

Primers used in this work are listed in Appendix A. For the construction of the *pteF* (gene ID: *GM000692*) and *pteR* (gene ID: *GM000694*) disruption mutant, two homologous fragments, ΔpteF-L (2034 bp) and ΔpteF-R (2076 bp), flanking the *pteF* operon were amplified from the genomic DNA of S0 via PCR using ΔpteF-LF/R and ΔpteF-RF/R as primer pairs; two homologous fragments, ΔpteR-L (2070 bp) and ΔpteR-R (1927 bp), flanking the *pteR* operon were amplified from the genomic DNA of S0 via PCR using ΔpteR-LF/R and ΔpteR-RF/R as primer pairs. The ΔpteF-L and ΔpteF-R PCR products were cloned into pKC1139 digested by EcoRI/XbaI using Gibson assembly to obtain pKC1139::ΔpteF; the ΔpteR-L and ΔpteR-R PCR products were cloned into pKC1139 digested by EcoRI/XbaI using Gibson assembly to obtain pKC1139::ΔpteR. Then, the resultant plasmid was introduced into strain S0 via conjugation to construct the *pteF* null mutant (savΔ*pteF*) and *pteR* null mutant (savΔ*pteR*) via homologous recombination as previously described [14]. The mutant strains savΔ*pteF* and savΔ*pteR* were confirmed via PCR with the primer pairs ΔpteFY-F/R and ΔpteFLY-F/R, as well as ΔpteRY-F/R and ΔpteRLY-F/R, respectively.

### 2.3. Transcriptome Analysis

*S. avermitilis* S0 and savΔ*pteF* were grown in liquid fermentation medium and collected at day 2 and 6 for RNA isolation. Triplicate samples of total RNA were extracted and purified for RNA library preparation and sequencing. Sequencing libraries were prepared using NEBNext^®^ Ultra™ Directional RNA Library Prep Kit for Illumina^®^ (NEB, Ipswich, MA, USA) and TruSeq PECluster Kit v3-cBot-HS (Illumina, Inc., San Diego, CA, USA) following the manufacturer’s recommendations before being sequenced on an Illumina HiSeq X-ten instrument in a paired-end (150 × 150 bp) approach (Novogene, Inc., Beijing, China). The high-quality reads were mapped using Bowtie/2.2.3 against the *S. avermitilis* S0 genome (GenBank accession code: GCA_029277425.1). HTSeq v0.6.1 was used to count the read numbers mapped to each gene. The fragment per kilobase of exon per million fragments mapped (FPKM) of each gene was calculated based on the length of the gene and the read count mapped to this gene, and statistically significant differences in gene expression were detected using the DESeq R package (1.18.0) in accordance with the criteria |log_2_ (fold change)| > 1.0 and *p*-value < 0.05. The technical and biological reproducibility of RNA-seq results was evaluated to be good via three independently fermented cDNA samples. Pearson correlation coefficients of the sequencing results were evaluated to be higher than 0.95 based on the log-transformed FPKM. Kyoto encyclopedia of genes and genomes (KEGGs) is a database resource for understanding high-level functions and utilities of the biological system (http://www.genome.jp/kegg/, accessed on 2023). ClusterProfiler (Version 3.8.1) software was used to test the statistical enrichment of differential expression genes in KEGG.

### 2.4. Extraction and High-Performance Liquid Chromatography (HPLC) Analysis of Avermectin B_1a_ and Filipin III

Avermectin B_1a_ was extracted and analyzed as described previously [11]. The procedure for the extraction of filipin III was the same as the method for avermectin B_1a_, and the concentration of filipin III was determined as previously described [27].

### 2.5. Genomic Synteny Analysis

Genomic alignment between the *S. avermitilis* S0 genome and *S. avermitilis* MA-4680 genome were performed using the MUMmer (Version 3.23) [28] and LASTZ (Version 1.03.54) [29] tools. Genomic synteny was analyzed based on the alignment results.

### 2.6. Statistical Analysis

All experiments were performed independently in at least three biological triplicates, and data were presented as mean values ± standard deviation (s.d.). Student’s *t*-test (two-tail) was used to demonstrate the statistical significance, with * *p* < 0.05, ** *p* < 0.01, *** *p* < 0.001, **** *p* < 0.0001 and “ns” meaning not significant.

## 3. Results

### 3.1. Analysis of Avermectin and Filipin Biosynthesis in S. avermitilis S0

*S. avermitilis* S0 is a high-yield avermectin-producing strain obtained via random mutation from *S. avermitilis* MA-4680 [11]. Genome analysis showed that there are two long inverted repeat segments in the flanking regions (left arm: 1 bp to 963,064 bp; right arm: 8,668,384 bp to 9,631,365 bp) in the S0 strain (Figure 1a). Each repeated segment contains six BGCs according to comparative genomic analysis between MA-4680 and S0, including three polyketide BGCs, *ave*, *pte* and *pks11* (Table 1), and they compete with acyl-CoA precursors while the known clusters *pte* and *ave* have same direct precursors, malonyl-CoA and methylmalonyl-CoA (Figure 1b). These data suggested that avermectin biosynthesis might significantly influenced by *pte* and *pks11* because they doubled in copy number. We first analyzed the time course transcription level of *pte* and *pks11* based on the transcriptome data on S0. It was shown that genes involved in *pte* were constantly transcribed at a level similar to that of genes in *ave* during the whole fermentation process (Figure 1c). Also, we detected filipin III, the major component of the filipin complex [30], in the liquid fermentation broth of S0 (Figure 1d). Given the fact that filipin causes competition between essential precursors and reducing power with avermectins, we thought it might be helpful to improve avermectin production in S0 by inhibiting filipin biosynthesis.

### 3.2. Influence of PteF and PteR on Avermectin and Filipin Production

Generally, deleting the activator of target BGCs is an effective approach to blocking the biosynthesis of corresponding metabolites. Therefore, in-frame deletion was carried out to obtain *pteF* and *pteR* deletion mutants savΔ*pteF* and savΔ*pteR*. Compared with the parent strain S0, the titer of filipin III in savΔ*pteF* and savΔ*pteR* decreased by 63.5% and 9.1%, respectively (Figure 2a), demonstrating that PteF plays a significant active role in filipin biosynthesis in S0, which agrees with previous investigations obtained from other *Streptomyces* [24,27]. We also found that titers of avermectin B_1a_ in savΔ*pteF* and savΔ*pteR* were, respectively, 55.49% and 26.12% higher than those of the parent strain, S00 (Figure 2b). Moreover, the deletion of *pteF* boosted avermectin biosynthesis during the whole production stage (day 4–10) (Figure 2a,b). These data demonstrate that PteF might regulate the biosynthesis of filipin and avermectin simultaneously. Although many investigations have demonstrated the regulatory mechanism of PteF on filipin biosynthesis, to date, it remains unclear how PteF influences avermectin biosynthesis.

### 3.3. Influence of pteF Deletion on Transcription of Genes Involved in Polyketide Synthase (PKS) BGCs

To comprehensively explore the influence of *pteF* deletion on avermectin biosynthesis, we carried out transcriptome data analysis on S0 and savΔ*pteF* in the exponential phase (day 2) and mid-production phase (day 6). Since many CSRs are involved in orchestrated regulatory networks controlling the biosynthesis of more than one secondary metabolite [19], we first focused on the transcription level changes in genes in putative BGCs encoding polyketides that might compete for precursors and reducing power with avermectins. We found that the deletion of *pteF* significantly reduced the transcription level of genes in *pte* on day 2 and 6 (Figure 3a,b), while *pteF* deletion had almost no effect on the transcription level of genes involved in other PKS BGCs on day 2 and 6 (Figure 3c,d). Despite that, we also found that the transcription levels of genes in *ave* were slightly increased (Figure 3c,d). Moreover, the FPKM value of genes in *ave* quite high compared with those in the other PKS BGCs, increasing by 4- to 500-fold and from 4- to 750-fold on day 2 and 6 in S0, respectively (Figure 3c,d and Appendix A). These data imply that, on one hand, PteF might not directly regulate the transcription of genes in PKS BGCs; on the other hand, the enhanced titer of avermectin B_1a_ might be due to a slight increase in the gene transcription level of *ave*, as well as the availability of more precursors and reducing power, which might be caused by the indirect effect of *pteF* deletion.

### 3.4. Influence of pteF Deletion on Global Regulatory Network

To further explore the indirect effect of *pteF* deletion on avermectin production, we conducted a comprehensive investigation of the transcriptome data. Considering that many CSRs are also global regulatory proteins, we further investigated the global influence of *pteF* deletion on the regulatory network. According to the transcriptome data, *pteF* deletion significantly influenced the transcription level of 141 genes on day 2, including 17 up-regulated and 124 down-regulated genes (Figure 4a and Appendix A), as well as 100 genes on day 6, including 38 up-regulated and 62 down-regulated genes in different metabolic pathways (Figure 4b and Appendix A). The data suggest that PteF not only tightly regulates filipin biosynthesis, but also exhibits a global regulatory effect on cell metabolism. We first focused on changed regulatory genes. *pteF* deletion resulted in 12 and 5 putative regulatory genes with significant transcription level changes on day 2 and 6, respectively (Figure 4b, Table 2 and Table 3). These genes encode sigma/anti-sigma factors and transcriptional regulators from various families. Among them, *GM003939* and *GM003574*, encoding putative anti-sigma regulatory factors, showed significant transcription level changes at both day 2 and 6. Transcription levels of *GM003939* were significantly increased, while those of *GM003574* were decreased (Table 2 and Table 3). Furthermore, we also found that *GM007052*, which encodes for a putative RNA polymerase extracytoplasmic function (ECF) subfamily sigma factor, showed significantly decreased gene transcription levels (Table 2). Since the expression of either sigma factors or anti-sigma factors influences the transcription of their potential target genes, we speculated that *pteF* deletion might have led to significant metabolic changes that would have favored avermectin biosynthesis.

### 3.5. Influence of pteF Deletion on Cell Metabolism

To explore the effect of *pteF* deletion on changes in metabolic pathways, KEGG enrichment analysis was carried out to explore the global changes between savΔ*pteF* and S0. Data showed that there were four and seven metabolic pathways with significant changes (gene ratio > 0.05, *p*-value < 0.05) on day 2 and 6, respectively (Appendix A). Among these genes, we first noted three metabolic pathways: glyoxylate and dicarboxylate metabolism, glycerolipid metabolism and propanoate metabolism pathway. These primary metabolic pathways are linked together by central carbon metabolism, and participate in the metabolism of different acyl-CoAs required for avermectin biosynthesis (Figure 5). Nine genes with transcription level changes were involved in three pathways. The transcription level of gene *acsA4* (*GM002244*), encoding putative acetyl-CoA synthetase, was increased on day 2 and 6 (Figure 5 and Appendix A), and that of *accA1* (*GM006049*), encoding putative acetyl/propionyl-CoA carboxylase alpha subunit, was also up-regulated on day 2. Such a change might favor the biosynthesis of acetyl-CoA, propionyl-CoA, malonyl-CoA, and methylmalonyl-CoA, which are required for avermectin biosynthesis (Figure 5). Moreover, we also found that *tgs* (*GM008560*) and *plsC4* (*GM007927*), which encode for putative triacylglycerol synthase and 1-acylglycerol-3-phosphate O-acyltransferase, respectively, and participate in triacylglycerol biosynthesis, showed down-regulated transcription levels after *pteF* deletion on day 6. These data imply that *pteF* deletion might reduce carbon flux towards triacylglycerols that compete with acyl-CoA precursors for avermectin production [14]. Meanwhile, we also found that *pteF* deletion led to a decreased transcription level of *acnA* (*GM002508*), which encodes an aconitate hydratase in the tricarboxylic acid (TCA) cycle on day 6 (Figure 5 and Appendix A). Since the repression of the TCA cycle has been proven to decrease carbon flux competition with polyketide biosynthesis [14], we speculated that the reduced expression level of TCA cycle caused by *pteF* deletion might also have been a possible reason for the improved avermectin production. In addition to genes in primary metabolic pathways, KEGG pathway enrichment analysis also showed genes involved in secondary metabolic pathways, such as *pteB* (crotonyl-CoA reductase) and *pteD* (cytochrome P450 hydroxylase) in *pte* (Appendix A). All of these data indicate that *pteF* deletion rearranged cell metabolism, possibly redirecting more carbon flux from primary metabolism to the avermectin biosynthetic pathway.

## 4. Discussion

Regulator engineering is a convenient strategy to increase the production of valuable natural products. Given the fact that natural product biosynthesis is tightly controlled by complicated cellular regulatory networks, it is necessary to elucidate the influence of key regulators on target chemical production prior to strain engineering [31]. Avermectins, as valuable pharmaceuticals, possess diverse applications. Although the industrialization of avermectin has been highly developed, its production still needs to be improved for lower application costs. In the avermectin-producing strain *S. avermitilis*, oligomycin and filipin are generally biosynthesized along with avermectins. Moreover, CSRs such as OlmR I and OlmR II in *olm*, AveR in *ave*, and PteF, and PteR in *pte*, showed cross-regulation relationships between oligomycin and filipin, as well as oligomycin and avermectin [22,32]. However, the influence of PteF and PteR in *pte* on avermectin biosynthesis has been scarcely investigated. Moreover, many CSRs also possess global regulatory functions; thus, we also aimed to comprehensively explore the influence of PteF and PteR. Uncovering the influences of these two regulators would have facilitated the strain engineering of avermectin producers for higher yields.

Gene inactivation experiments showed that PteF plays a more important role than does PteR in regulating the biosynthesis of both filipin and avermectin B_1a_. Therefore, we further focused on the influence of PteF in *S. avermitilis* S0. Based on time course transcriptome data analyses, we found that PteF showed a small influence on putative PKS BGCs including *ave*, whereas *pteF* deletion resulted in significant changes in many proteins closely related to gene transcription, including transcription regulators and sigma/anti-sigma factors. Among them, *GM007052* and *GM003939,* with the most significant transcription level changes on day 2 and 6, encoded a sigma factor and an anti-sigma factor, respectively. The sigma factor, as an essential component of bacterial RNA polymerases, determines promoter specificity and regulates transcription as well as translation, while anti-sigma factors are responsible for inhibiting sigma factor function. In *Streptomyces*, sigma factors generally play essential roles in regulating signal transduction and regulatory networks, as well as secondary metabolite biosynthesis to assist their survival in complex environments [33]. For instance, the housekeeping σ factor σ*^hrd^^D^* regulates the biosynthesis of actinohordin and undecylprodigiosin in *Streptomyces coelicolor* [34], and promotes avermectin biosynthesis in *S. avermitilis* [8]. In addition to σ*^hrdB^*, σ^6^ [35], σ^25^ [36], and σ^8^ [37] in *S. avermitilis* have been reported as negative regulators of avermectin biosynthesis. Here, we found that 6 sigma/anti-sigma factors and 11 regulators in total were significantly influenced, implying that PteF might also act as a global regulator influencing cellular regulatory networks and favoring avermectin biosynthesis. However, the function of these regulatory proteins has never been characterized before. Therefore, further investigations on these regulators might favor regulatory network engineering for a higher yield of avermectins.

Changes in the regulatory network will generally influence cellular metabolic flux changes. Data analysis demonstrate that improved avermectin production might be due to the inhibition of filipin biosynthesis, repression of the TCA cycle, and improved supply of avermectin biosynthetic precursors. In addition to these findings, we further speculated that PteF might influence nitrogen metabolism and metabolic flux from primary metabolism to secondary metabolism. KEGG pathway enrichment analysis showed that *pteF* deletion led to a down-regulation of nitrogen metabolism in S0 on day 2 (Appendix A). Coincidentally, several proteins that are closely related to nitrogen metabolism were transcriptionally repressed on day 2, including the putative ECF sigma factor GM001189 as well as its upstream putative urea ABC transporter (GM001184-GM001187), a putative nitrate extrusion protein NarK (GM005862), a putative nitrite reductase (NAD(P)H) large subunit NirB (GM006513), a putative glutamine synthetase GlnA2 (GM006947), the urease gamma subunit UreA (GM008344) and beta subunit UreC1 (GM008348) (Appendix A). Since nitrogen sources are essential for the biosynthesis of protein and nucleic acid required for biomass, the repressed nitrogen metabolism on day 2 caused by *pteF* deletion might have reduced cell growth and saved cellular energy for other physiological activities, thus redirecting more carbon flux toward secondary metabolism, which is also in line with previous findings that nitrogen metabolism influences production of secondary metabolites in *Streptomyces* [38].

## 5. Conclusions

Our work investigated the relationship between two important BGCs, *pte* and *ave*, with two copies in *S. avermitilis* S0. Moreover, we comprehensively explored the influence of PteF on avermectin biosynthesis, as well as its influence on global regulatory and metabolic networks. We discovered many candidate targets with potential engineering value based on transcriptome data. As most of them were previously unreported, more experimental validations are required for detailed characterization. Despite that, our work gave new insight into regulatory networks regulating avermectin biosynthesis, which might provide more guidance on the further strain engineering of *S. avermitilis* for obtaining a high yield of avermectin.

## Figures and Tables

**Figure 1 biology-13-00344-f001:**
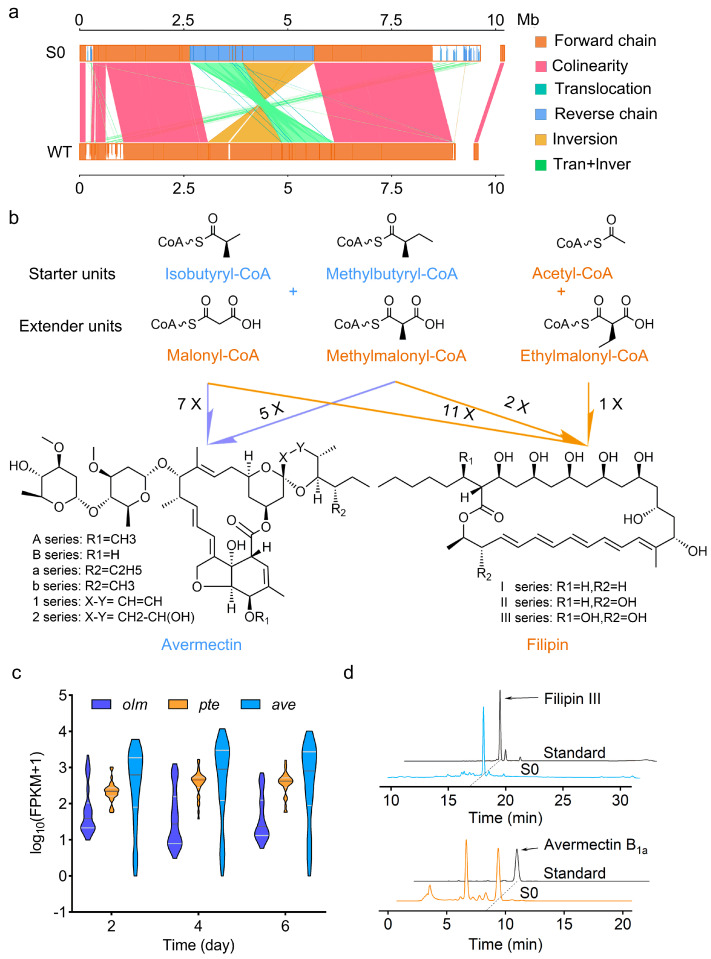
Analysis of avermectin and filipin biosynthesis in *S. avermitilis* S0. (**a**) Synteny analysis of S0 and MA-4680. (**b**) Acyl-CoA precursors for avermectin and filipin biosynthesis. Shared precursors are highlighted in orange. (**c**) Transcription level analysis of genes in *ave*, *olm* and *pte*. Data were obtained from transcriptome data on S0 at 2, 4, and 6 days from three independent repeats. *pte*, *olm* and *ave* indicate BGCs for filipin, oligomycin, and avermectin, respectively. (**d**) Analysis of avermectin and filipin of S0.

**Figure 2 biology-13-00344-f002:**
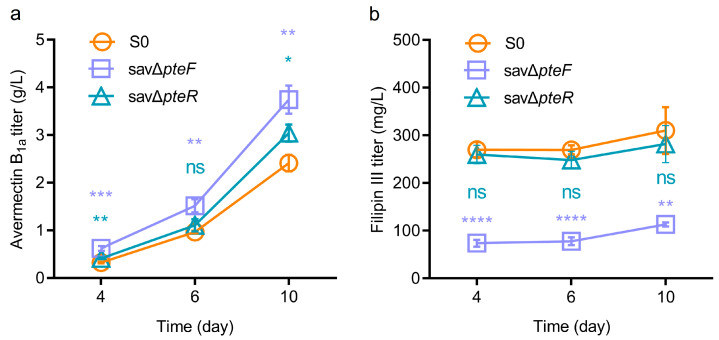
Effect of *pteF* and *pteR* deletion on avermectin and filipin production. (**a**) Filipin III titer of mutant strain savΔ*pteF* and savΔ*pteR* under different fermentation times. (**b**) Avermectin B_1a_ titer of mutant strain savΔ*pteF* and savΔ*pteR* under different fermentation times. Data were obtained from three independent replicates. Student’s *t*-test (two-tail) was used to demonstrate the statistical significance, with * *p* < 0.05, ** *p* < 0.01, *** *p* < 0.001, **** *p* < 0.0001 and “ns” meaning not significant.

**Figure 3 biology-13-00344-f003:**
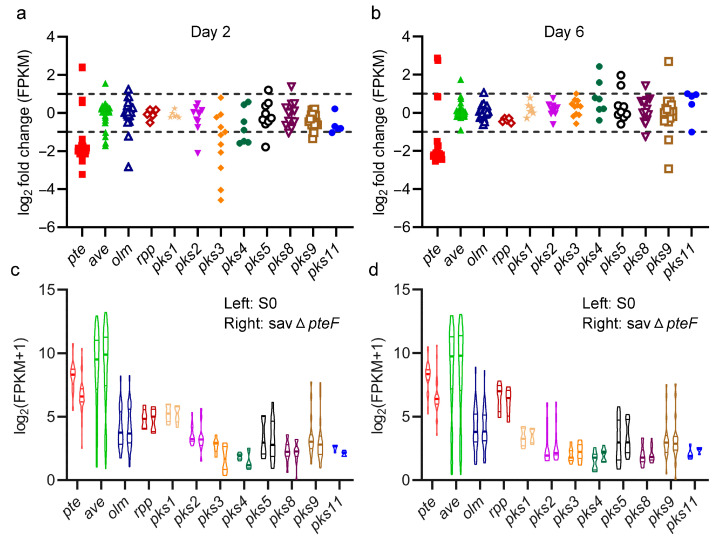
Effect of *pteF* deletion on transcription level changes in genes in different polyketide BGCs. Relative changes in transcriptional level of genes in polyketide BGCs on day 2 (**a**,**c**) and day 6 (**b**,**d**). The points in (**a**,**b**) represent the log_2_-transformed fold change of FPKM for each gene, indicating the variation in expression levels of individual genes within the PKS gene cluster. The violin plots in (**c**,**d**) show the log_2_-transformed of (FPKM + 1) for all genes within the cluster, representing the transcriptional level differences across the entire gene cluster between the S0 and savΔ*pteF* strains. The dashed lines in (**a**,**b**) indicate the threshold of the genes with significant changes.

**Figure 4 biology-13-00344-f004:**
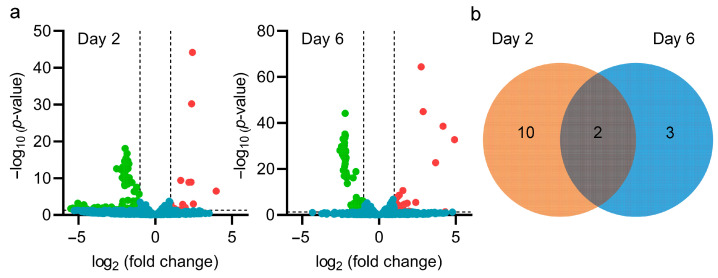
Global influence of *pteF* deletion on gene transcription level in S0. (**a**) Volcano plots of significantly different genes on day 2 and 6. The red, green, and blue point represent genes with significantly up-regulated (log_2_ (fold change) > 1, *p*-value < 0.05), significantly down-regulated (log_2_ (fold change) < −1, *p*-value < 0.05), and scarcely changed profiles, respectively. (**b**) Venn diagram of significantly changed regulator genes on day 2 and 6.

**Figure 5 biology-13-00344-f005:**
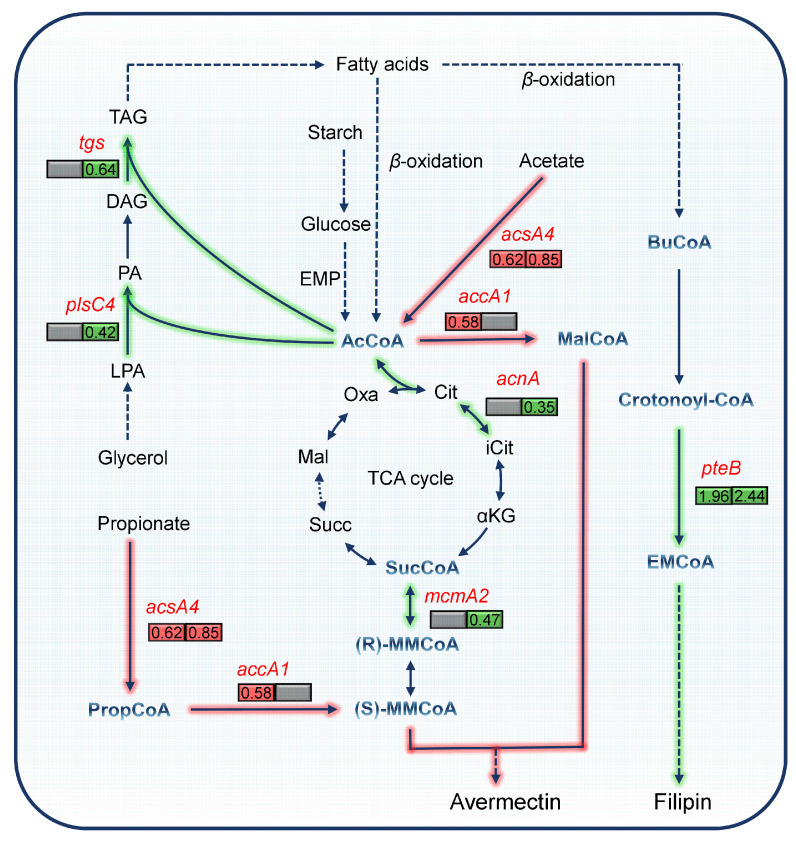
Effect of *pteF* deletion on cell metabolism. Genes enriched in significantly different metabolic pathways were mapped to a precursor supply metabolic pathway of avermectin biosynthesis. The red, green, and gray colors represent genes with up-regulated, down-regulated, and scarcely changed transcription levels at day 2 (left) and day 6 (right), respectively. The data in the frame represent the absolute values of the log_2_-transformed value (fold change) for the corresponding genes. Solid arrows represent a one-step reaction, while dashed arrows represent multi-step reactions. The red and green arrow indicate the pathway with enhanced and decreased transcription level, respectively. Abbreviations: AcCoA, acetyl-CoA; MalCoA, malonyl-CoA; PropCoA, propionyl-CoA; MMCoA, methylmalonyl-CoA; SucCoA, succinyl-CoA; BuCoA, butyryl-CoA; EMCoA, ethylmalonyl-CoA; TAG, Triacylglycerol; DAG, 1,2-Diacylglycerol; PA, Phosphatidic acid; LPA, Lysophosphatidic acid; Oxa, Oxaloacetate; Cit, Citrate; iCit, Isocitrate; αKG, 2-Oxoglutarate; Succ, Succinate; Mal, Malate.

**Table 1 biology-13-00344-t001:** BGCs involved in each repeated segment.

Category	Cluster	Length	Product
terpene	*ams*	1008 bp	Avermitilol, avermitilone
polyketide	*pks11*	5227 bp	Polyketide ^1^
other	*mcj1*	3368 bp	Microcin ^1^
polyketide	*pte*	80,344 bp	Filipin
polyketide	*ave*	80,849 bp	Avermectin
terpene	*crt*	8718 bp	Isorenieratene

^1^ Putative products.

**Table 2 biology-13-00344-t002:** Significantly changed regulator genes on day 2.

Gene ID	Functional Description	log_2_ (Fold Change)	*p*-Value
*GM007052*	putative RNA polymerase ECF subfamily sigma factor	−5.05	0.001
*GM006412*	putative IclR family transcriptional regulator	−2.29	0.020
*GM003939*	putative anti-sigma regulatory factor	2.19	1.42 × 10^−9^
*GM000222*	putative TetR family transcriptional regulator	−1.68	0.022
*GM007217*	putative two-component system response regulator	1.57	0.042
*GM003574*	putative anti-sigma regulatory factor	−1.51	1.46 × 10^−9^
*GM003618*	AsnC family transcriptional regulator	−1.47	0.044
*GM001189*	putative RNA polymerase ECF subfamily sigma factor	−1.30	0.037
*GM000932*	putative LacI family transcriptional regulator	−1.24	0.049
*GM005356*	putative two-component system sensor kinase	−1.06	0.001
*GM007707*	putative MerR family transcriptional regulator	1.02	0.036
*GM001883*	putative ArsR family transcriptional regulator	−1.00	0.030

**Table 3 biology-13-00344-t003:** Significantly changed regulator genes on day 6.

Gene ID	Functional Description	log_2_ (Fold Change)	*p*-Value
*GM003939*	putative anti-sigma regulatory factor	4.17	2.51 × 10^−39^
*GM003307*	putative regulatory protein	1.95	0.024
*GM008359*	LytR family transcriptional regulator	1.23	0.037
*GM003574*	putative anti-sigma regulatory factor	−1.13	6.45 × 10^−7^
*GM002997*	TetR family transcriptional regulator	−1.01	3.05 × 10^−4^

## Data Availability

All data generated or analyzed during this study are included in this manuscript and its Appendix A. Requests for any additional information can be made to the corresponding authors.

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
