# Peer review of "Influence of Cluster-Situated Regulator PteF in Filipin Biosynthetic Cluster on Avermectin Biosynthesis in Streptomyces avermitilis"

_biology, 2024, doi:10.3390/biology13050344_

Round 1
Reviewer 1 Report
Comments and Suggestions for Authors
Summary
The manuscript submitted by Du and coworkers describes a study focused on investigating the function of two regulatory factors, PteF and PteR, present within the filipin gene cluster in Streptomyces avermitilis. Specifically, they investigate the effect of deleting these genes on the production of avermectin using metabolomic and transcriptomic analyses. They find that the deletion of PteF results both a reduction in filipin production and a significant increase in avermectin production. They attribute the increase in avermectin titer to the role of PteF as a global regulator, the deletion of which favors metabolite flux towards avermectin biosynthesis.
General Comments
The observation that avermectin production is enhanced by the deletion of PteF is an interesting one and provides avenues to increase avermectin titers in commercial strains. The upregulation of a number of genes in the PteF knock-out mutant supports the author’s conclusion that PteF is a global regulator, but the explanation as to why avermectin production increases is somewhat hand-wavy. Metabolomic data in combination with the transcriptomic data collected would solidify the author’s conclusions and provide a more comprehensive picture with regards to metabolic flux, although this may be beyond the scope of this study.
Specific Comments
Line 360: ‘our work gave new insight into regulatory and metabolic networks regulating avermectin biosynthesis’ – There’s a lack of metabolic data in this manuscript so I would rephrase this sentence in the conclusion to focus only on the regulatory aspect.
Author Response
Point-by-point response:
Reviewer #1:
Summary
The manuscript submitted by Du and coworkers describes a study focused on investigating the function of two regulatory factors, PteF and PteR, present within the filipin gene cluster in Streptomyces avermitilis. Specifically, they investigate the effect of deleting these genes on the production of avermectin using metabolomic and transcriptomic analyses. They find that the deletion of PteF results both a reduction in filipin production and a significant increase in avermectin production. They attribute the increase in avermectin titer to the role of PteF as a global regulator, the deletion of which favors metabolite flux towards avermectin biosynthesis.
General Comments
The observation that avermectin production is enhanced by the deletion of PteF is an interesting one and provides avenues to increase avermectin titers in commercial strains. The upregulation of a number of genes in the PteF knock-out mutant supports the author’s conclusion that PteF is a global regulator, but the explanation as to why avermectin production increases is somewhat hand-wavy. Metabolomic data in combination with the transcriptomic data collected would solidify the author’s conclusions and provide a more comprehensive picture with regards to metabolic flux, although this may be beyond the scope of this study.
Response:
Thanks for your valuable suggestions.
Currently, we speculated that the improved titer of avermectin might be ascribed to the increased metabolic flux towards precursors for avermectin biosynthesis based on transcriptome data, while as you suggested, metabolomic data in combination with the transcriptomic data would solidify the conclusions. In future work, we will further investigate the impact of PteF inactivation on metabolic flux by analyzing metabolomics and isotopic data, providing guidance for the engineering of high-yield avermectin-producing Streptomyces avermitilis.
Specific Comments
Line 360: ‘our work gave new insight into regulatory and metabolic networks regulating avermectin biosynthesis’ – There’s a lack of metabolic data in this manuscript so I would rephrase this sentence in the conclusion to focus only on the regulatory aspect.
Response:
Thanks for your valuable comments.
As you suggested, we have revised this sentence in the new version of manuscript (Page 11, Lines 364-365).

Reviewer 2 Report
Comments and Suggestions for Authors
Crosstalk regulation is important in secondary metabolism of Streptomyces species. Avermectins are important green natural product pesticides. Guozhong Du and co-authors focused on interesting transcription factors PteF and PteR. Firstly, PteF and PteR knockouts were first employed in this study to obtain high-yield strains of avermectins, particularly the PteF knockout strain. Subsequently, transcriptional analysis was conducted on the Streptomyces PteF knockout strain, investigating the global transcriptional effects of PteF. This work provided new targets of the regulatory and metabolic networks that potentially benefit avermectin biosynthesis. However, I think the manuscript should make some minor revisions to get improvement before acceptance:
1. How many biological repeats are there in Figure 2? Especially Figure 2a, savΔpteR, seems like no error bar. It's possible that the error bars are very small or not visible in the plot. Change into a different color maybe helpful.
2. Can you explain why pteR deletion has no effect on filipin production, but increase avermectin B1a production?
3. Can you provide the sequence information of pteF and pteR, such as Gene ID or DNA/protein sequence? So that researchers interested in those genes can get sequence easily.
4. It is recommended to show critical gene transcription changes in Figure 5 cell metabolism network, which are currently present in the supporting information.
5. Please define ABBRs throughout the manuscript. Such as “SAPR”, “KEGG”, etc.
6. The manuscript has mixture written of “pteF/pteR deletion” and “PteF/PteR deletion”. The term "deletion" refers to a gene. Please carefully check throughout the manuscript and unify all instances to “pteF/pteR deletion”.
Author Response
Point-by-point response:
Reviewer #2: Crosstalk regulation is important in secondary metabolism of Streptomyces species. Avermectins are important green natural product pesticides. Guozhong Du and co-authors focused on interesting transcription factors PteF and PteR. Firstly, PteF and PteR knockouts were first employed in this study to obtain high-yield strains of avermectins, particularly the PteF knockout strain. Subsequently, transcriptional analysis was conducted on the Streptomyces PteF knockout strain, investigating the global transcriptional effects of PteF. This work provided new targets of the regulatory and metabolic networks that potentially benefit avermectin biosynthesis. However, I think the manuscript should make some minor revisions to get improvement before acceptance:
Point 1: How many biological repeats are there in Figure 2? Especially Figure 2a, savΔpteR, seems like no error bar. It's possible that the error bars are very small or not visible in the plot. Change into a different color maybe helpful.
Response:
Thanks for your careful review.
Data were obtained from three independent replicates in Figure 2. We have added the explanation of the biological repeats in the figure legend (Page 6, Line 205-206). Meanwhile, we have revised the Figure 2 to make the error bar more clearly (Page 6, Line 202).
Point 2: Can you explain why pteR deletion has no effect on filipin production, but increase avermectin B1a production?
Response:
Thanks for your question.
PteR and PteF are two regulators involved in the filipin biosynthetic gene cluster. Our results (Figure 2) showed that PteF plays a major role in controlling the biosynthesis of filipin in pte. We speculated that PteR might also influence filipin production in the absence of PteF. Despite pteR deletion in our work shown no effect on filipin production, similar to PteF, PteR might also influence other metabolic pathways, which might be the reason why pteR deletion increased avermectin B1a production. However, currently in the present work, we did not focus on PteR and we do not have more data to support our speculation.
Point 3: Can you provide the sequence information of pteF and pteR, such as Gene ID or DNA/protein sequence? So that researchers interested in those genes can get sequence easily.
Response:
Thanks for your careful review.
The gene ID of pteF and pteR are GM000692 and GM000694 in Streptomyces avermitilis S0 (genome accession no. CGMCC 4.8011), respectively. We have provided the sequence information on page 3, lines 112-113.
Point 4: It is recommended to show critical gene transcription changes in Figure 5 cell metabolism network, which are currently present in the supporting information.
Response:
Thanks for your suggestion.
We have provided more data and revised Figure 5 as you suggested (Page 9, Line 290).
Point 5: Please define ABBRs throughout the manuscript. Such as “SAPR”, “KEGG”, etc.
Response:
Thanks for your careful review.
We have added defines for all abbreviations in the manuscript (Page 2, Line 68; Page 3, Lines 143 & 147; Page 6, Line 207; Page 7, Line 247).
Point 6: The manuscript has mixture written of “pteF/pteR deletion” and “PteF/PteR deletion”. The term "deletion" refers to a gene. Please carefully check throughout the manuscript and unify all instances to “pteF/pteR deletion”.
Response:
Thanks for your careful review.
We have changed all the mixture written into “pteF/pteR deletion” in the revised manuscript.
